**Data Availability Statement:** All data supporting our work is provided within the manuscript.

**Funding:** The study was supported by the South African Medical Research Council. DSI-NRF Centre

# A qualitative study of risks and protective factors against pregnancy among sexually-active adolescents in Soweto, South Africa

**Edna N. Bosire** [1,2,3‡] *, **Katharine Chiseri** [4‡], **Dawn L. Comeau** [4], **Linda Richter** [1], **Aryeh D. Stein** [2,5], **Shane A. Norris** [1,2,5]

1 DSI-NRF Centre of Excellence in Human Development, University of the Witwatersrand, Johannesburg, South Africa, 2 Department of Paediatrics, SAMRC Developmental Pathways for Health Research Unit, University of the Witwatersrand, Johannesburg, South Africa, 3 Center for Innovation in Global Health, Georgetown University, Washington, DC, United States of America, 4 Department of Behavioral, Social, Health Education Sciences, Rollins School of Public Health, Emory University, Atlanta, GA, United States of America, 5 Hubert Department of Global Health, Rollins School of Public Health, Emory University, Atlanta, GA, United States of America

‡ These authors are joint first authorship on this work.
* edna.bosire@wits.ac.za

## Abstract

Risky sexual behaviors contribute to increased risk of adolescent pregnancy. This qualitative study sought to understand risks and protective factors against pregnancy amongst sexually-active adolescents in Soweto, South Africa. We used purposive sampling to recruit women at age 24 years from Soweto, who self-reported having sexual debut by age 15 years. Twenty women were recruited: (i) women who did not become pregnant before 18 years (n = 10) and (ii) women who became pregnant before 18 years (n = 10). In-depth interviews were conducted to understand their family backgrounds, conversations about sex, sexual behaviors, and initiatives taken (or not) during adolescence to prevent pregnancy. Both groups of women reported predisposing risks to early pregnancy including influence from peers to engage in early sex, unstable family relationships and limited conversations about sex. We found that the family is a key institution in supporting adolescents' decisions regarding their behaviors and choices, as are peers and exposures to information. Community Youth Centers, high schools and Youth Friendly Health Services should ensure that adolescents have access to relevant information, including sex education and contraceptives.

## Introduction

Preventing unintended pregnancies and early childbirth amongst adolescents is a focus of the Sustainable Development Goals (SDG) [1]. The number of adolescent pregnancies has declined in most developed high-income regions but remains a major concern in low- and middle-income countries (LMICs). Every year, an estimated 21 million young women aged 15–19 years in developing countries become pregnant, of which almost half (49%) are

of Excellence in Human Development at the University of the Witwatersrand, Johannesburg, South Africa and the Gates Foundation supported the Birth to Twenty Plus cohort. The funders had no role in study design, data collection and analysis, decision to publish, or preparation of the manuscript.

**Competing interests:** The authors declare that they have no competing interests.

unintended [2], and more than half of the unintended pregnancies end in induced abortions or miscarriages [3]. Nearly half of induced abortions in developing countries (in regions of Africa, Asia, Latin America and the Caribbean) occur among adolescent girls and young women under the age of 25 years [4]. Adolescent mothers are also likely to be stigmatized, drop out of school and get married or die from unsafe abortion [5, 6], and could potentially have poor growth and development including that of their children–especially as their infants are more likely to be born premature and to die in the perinatal period [7].

South Africa is home to approximately 9.7 million adolescents [8]. The adolescent pregnancy rate is 47 births per 1000 females aged 15–19 per annum [9]. A study conducted in Soweto found that 23% of pregnancies carried by 13–16 year old young women and 14.9% in the 17–19 year age range are aborted [10]. Early sexual debut is common in South Africa and has been associated with increased rate of adolescent pregnancy and risk of HIV infection [11]. Yet, not all women who experience early sexual debut become pregnant. Therefore, understanding how some sexually active young women avoid pregnancy is important to reduce the adolescent pregnancy rate.

Predisposing factors for adolescent pregnancy include early marriage, unequal power relations and forced sexual initiation, gender-based violence, lack of skills to negotiate safer sex options [12], as well as lack of comprehensive sexuality education [13]. Poor parental supervision and single motherhood (due to parental separation) [14], unstable families, poor mother-daughter relationships, limited conversation between parents and adolescents on issues around sex [10] have also been associated with adolescent pregnancy. Sexual risk-taking behaviors, including unprotected sex, multiple sex-partners, low contraceptive use and negative peer pressure are common among young people in South Africa [15]. These factors have also been reported in other low-and middle-income countries (LMIC) [16].

A key strategy to reduce risk factor exposure has been sex education to empower adolescents [17]. Sex education programs can lead to improved knowledge and better adolescent reproductive health outcomes–including the ability to make informed and crucial choices [13]. In addition, economic empowerment can improve women's decision-making by changing the power structure in relations, and may enable access to contraceptives [18]. Other strategies involve engaging families, peers, healthcare services and schools to support and help adolescents to make decisions that are in their best interest, particularly as they transition from childhood to adulthood [19].

Studies on sexual and reproductive health of adolescents have tended to focus on vulnerabilities and exposure to various health and developmental risks, including unwanted pregnancies [19, 20]. However, despite being exposed to various risk factors, some adolescent girls are able to prevent unwanted pregnancies [21].

This qualitative study sought to understand risks and protective factors against pregnancy amongst sexually-active adolescents in Soweto, South Africa. The specific aim of this study was to explore how some adolescent girls in Soweto who experience early sexual debut–defined for the purpose of this study as engaging in their first sexual experience at or before age 15 years [22]–are able to successfully prevent pregnancy through age 18 years. The conceptualization of the study is innovative in that we utilized longitudinal data from adolescence to young adulthood to identify participants utilizing prospectively-collected data on sexual debut and pregnancy prevention.

## Materials and methods

### Study design

This was an exploratory study that used in-depth interviews with women in Soweto to understand risks to early pregnancy and factors that protected them during their adolescence period.

We explored issues such as conversations about sexual activity between adolescents and their parents/caregivers, partners, friends/peers, and school; condom and contraceptive use, as well as, steps that adolescents took, or didn't take, to prevent early pregnancies.

## Setting

This study was conducted in Soweto, a predominantly low-income, urban setting in Johannesburg with a population density of 6357 people per km$^2$ according to SA's most recent national census [23]. Soweto is a historically disadvantaged township near Johannesburg, with a long history of engagement in the anti-*Apartheid* struggle [24]. In Soweto, a third of women have their first child by the time they are 19 years old [25].

Birth to Twenty plus (Bt20+) is South Africa's longest existing longitudinal birth cohort, involving children born in Soweto-Johannesburg within a 7-week period between late April and early June 1990 [26]. The original cohort included 3273 mothers and their children, who have since been followed up for close to three decades. Those enrolled in Bt20+ had continued residence within the metropolitan area of Johannesburg-Soweto for at least 6 months after birth. The cohort was recruited from antenatal and public health facilities. This study was nested within the Bt20+ cohort.

## Eligibility and participant recruitment

Purposive sampling was used to compile a list of eligible participants from the larger Bt20 + cohort. Women were eligible to participate if they currently lived in Soweto and had self-reported sexual debut by age 15 years, as self-reported in earlier waves of the study. We abstracted information on whether cohort participants had a pregnancy prior to age 18 years and sampled within these strata. Of 64 eligible women, one participant withdrew after initial consent, due to the content of the study. This resulted in 63 eligible participants, in which 24 were identified in our records as not having been pregnant before age 18 years (group 1) and 39 did become pregnant before age 18 years (group 2). Participants were selected using a random number generator, five from each group at a time, and invited to participate.

## Development of interview guide

The interview guide and questions were developed according to the following domains and themes: (i) adolescent's living environment; (ii) sexual history, condom and contraceptive use; (iii) conversations with family members, peers, and schools about sex; and (iv) steps taken to prevent pregnancy. A pilot study using a focus group discussion (FGD) methodology was conducted with the Bt20+ female research staff (black Africans), at the Developmental Pathways for Health Research Unit, who understood the local languages and the context where the study was conducted. These staff were chosen based on their cultural competency and sensitivity of the topic. Staff members helped in the development of the interview guide, reviewed the questions and provided feedback on the language of interview questions and cultural appropriateness of topics. Subsequent changes to the interview guide were made to improve the natural flow of the conversation and to allow for more probing questions. Interview guide was adjusted throughout data collection to incorporate questions about emerging topics [27]. The final interview guide that was used is provided as [**S1 File**].

## Data collection

Data collection took place between June and July 2014, when participants were 24 years old. Participants were informed about the study objectives, and only those who voluntarily agreed

to participate were allowed to sign the consent form. Data were collected by KC through individual semi-structured interviews at the Bt20+ research unit in Soweto. Ten interviews were conducted with participants in each group, for a total of 20 interviews; this was sufficient to reach saturation with no new themes emerging. Each interview lasted approximately 45–60 minutes. Interviews were conducted in English (as all participants were able to speak in English). However, participants were allowed flexibility to express themselves in other local languages. A female, multilingual research assistant was always available during the interviews in case of need for translation and assisted in note taking. However, none of the participants used local languages during the interviews. Additional information collected during the interviews included women's living environment and information about their sexual history. Following each interview, the interviewer wrote extensive field notes, highlighting key issues that emerged after each interview. Participants were reimbursed for travel expenses (R150, equivalent to USD10), and were provided refreshments during the interview.

## Researcher characteristics and reflexivity

KC, a young American woman conducted the interviews with the 20 participants, asking culturally sensitive topics. Being an 'outsider' (non-South African) may have influenced her views on issues around sex and sexual behavior. To mitigate this, KC worked closely with Bt20 + research staff, who helped in developing questions and reviewing the interview guide to ensure questions were culturally appropriate. KC also engaged a multilingual research assistant who was always available in case of need for translations. In addition, a constant thoughtful process in reviewing field notes and interviews with the participants and other researchers involved in the study allowed flexibility in data collection, analysis, and reporting of study findings.

## Ethics statement

Written informed consent was obtained from the study participants after reading out the content of the information sheet and explaining the purpose of the study. Given the sensitivity of the topics that were discussed, interviews were conducted in a private room at the research unit in Soweto. Participant names were replaced with unique codes, and all information including consent forms were stored in a locked cupboard in the research unit. The study received approval by the University of the Witwatersrand Ethics Committee [M140481] and Emory University Institutional Review Board [IRB00073568].

## Data analysis

All interviews were audio-recorded and transcribed verbatim. Participant's names were anonymized during transcriptions. KC began preliminary data analysis while conducting interviews. After reviewing three transcripts and field notes, she developed a code tree, where she created a list of deductive and inductive codes from the data. The deductive codes came from topics in the interview guide and inductive codes captured new themes that emerged in the data [28]. She then developed a codebook, which included definitions, perceptions and adolescent experiences. This codebook was reviewed by two experts in the research team, and revisions were made where necessary. The final codebook, together with transcripts, were imported into Max Qualitative Data Analysis (MAXQDA10) qualitative software where coding was done. KC and ENB coded the data, new codes were added as they emerged from the data to a point where no new theme emerged. They then summarized the key themes and subthemes, which were reviewed by other researchers involved in the study. Identified discrepancies were discussed, resolved and reviewed further until consensus was reached. This led to

development of a final data classification. Thematic analysis was used to identify themes and patterns in the data. Key emerging themes included sexual activity and risks, conversations about sex, risk factors for early pregnancy and protective factors that prevented adolescents from getting pregnant.

## Results

Table 1 shows demographic characteristics of participants. All participants were Black Africans, 24 years old at the time of the interview, and self-reported age of sexual debut by 15 years of age. Many participants lived with extended family members as primary caregivers. Only one participant reported living with both parents. Completion of secondary school was more common among those who did not become pregnant before 18 years (group 1).

The results were examined by looking at overall themes and further examining differences between the two groups of young women. Although both groups were predisposed to similar risk factors to becoming pregnant, results show that participants who did not become pregnant by age 18 years had strong family support systems, were exposed to conversations about sex, and had access to information, which helped them to make decisions and choices such as consistent condom use that protected them from early pregnancies. We present our results along the four key themes that emerged from this study.

### Sexual activity and risks

We explored participants' sexual activities in terms of sexual history, the number of partners they had, and the risks associated with their sexual behaviors. This section provides combined findings from group 1 and group 2 participants. Firstly, we found that adolescents engaged in early sexual initiation due to negative influence from peers or friends:

'We were influencing each other to break our virginity, lots of things. We just talked about having sex, how it feels and all of that.' [Grp. 2. Participant 219]

**Table 1. Participant's socio-demographic characteristics.**

|  | Total number of Participants (N = 20) | Group 1: Did not become pregnant (N = 10) | Group 2: Became pregnant (N = 10) |
|---|---|---|---|
| **Area of residence during adolescence** |  |  |  |
| Soweto | 18 | 9 | 9 |
| Outside of Soweto* | 2 | 1 | 1 |
| **Primary Caregiver(s) during adolescence**** |  |  |  |
| Both parents | 1 | 1 | 0 |
| Both parents and siblings | 3 | 2 | 1 |
| Mother | 4 | 2 | 2 |
| Mother and siblings | 3 | 1 | 2 |
| Grandparents | 3 | 2 | 1 |
| Other relatives (aunt/uncle/cousins/siblings) | 6 | 2 | 4 |
| **Completed high school** |  |  |  |
| Yes | 13 | 9 | 4 |
| No | 4 | 0 | 4 |
| Unknown | 3 | 1 | 2 |

* Participants who reported not living in Soweto during adolescence attended boarding school outside of Soweto or temporarily living in another province.

** Participants reported who they lived with as an adolescent—this person or these people were categorized as 'Primary Caregiver(s)'.

Three participants from both groups reported that they did not particularly want to engage in sex but agreed to do so with a partner who was pushy or who told them that they 'would be fine' as exemplified below:

*'He was older than me. He knew everything about sex, and I was not ready yet. He asked me; 'can we have sex' and I said I was scared but I said 'yes." (Grp. 1. Participant 118)*

Secondly, both groups had a small number of participants (n = 3) who had been in long-term relationships with their partners, so they had talked about having sex for quite some time and had prepared for that moment:

*'We talked about having sex because we started dating in grade 9. So, along the years we have been talking about it.' (Grp. 2. Participant 237).*

In addition, two participants from group 1 revealed that they strictly had one sexual partner; *'I'm so strictly with one partner, because of the preaching of my pastor' [Grp. 1. Participant 114]*. Participants revealed that having one sexual partner and being in a long term relationship was advantageous as it created a conducive environment for adolescents and their partners to discuss about consistent condom use.

More participants from group 1 reported that they consistently used contraceptives (or what they largely referred to as 'protection')–mostly condoms and birth control pills;

*'She [elder sister] gave me birth control pills, saying 'No you can take this' you won't get pregnant if the condom broke.' [Grp. 1. Participant 118]*

Thirdly, participants from both groups described various risky sexual behaviors that themselves or their friends/peers engaged in. These included inconsistent and improper condom use, having multiple sex partners and alcohol abuse:

*'When it's dark when the lights are off, they will pretend as if they are putting the condom on, then they will take it off.' [Grp. 2. Participant 233]*

*'My other cousins told me how she sleeps around and has many boyfriends.' [Grp. 1. Participant 117].*

Four participants from both groups mentioned having used alcohol before sex: *'No it [sex] just happened we were drunk.' [Grp. 1. Participant 105]*. Alcohol use and sex were also reported amongst their peers, close family members, especially their cousins, which promoted risky sexual behaviors:

*"She [cousin] would say 'I was out with my boyfriend drinking alcohol, having fun, enjoying and having sex.'" [Grp. 2. Participant 235]*

## Conversations about sex

We also investigated whether conversations about sex–between adolescents and their parents/caregivers, partners, friends/peers, and with teachers at school, occurred before or after the participant's sexual debut, and how this influenced her sexual behavior.

**Group 1 participants (those who did not become pregnant before 18 years).** Eight participants in group 1 reported that they had conversation about sex including other topics such

as pregnancy prevention methods (e.g., use of condoms), peer pressure, and about HIV/STI. Conversations largely occurred before sexual debut, and between adolescents and their mothers (whether mother was a primary caregiver or not), followed by relatives including aunts and grandmothers and with friends:

> 'Yes, oh yes. . .. I learned not to fall pregnant at an early stage, and also how to prevent diseases, I never had those diseases, like vaginal diseases or whatever.' [Grp. 1. Participant 103]

In this context, the nature of conversation was approached with a more open mindset and more of guiding and advising:

> '[mother] was educating me a lot about sex about using protection, diseases, all the things like sexual intercourse and pregnancy'. [Grp. 1. Participant 103]

> "At age 15, [My] aunt started the conversation. She emphasized that 'boys will give you babies; you have to be careful.'" [Grp. 1. Participant 108]

Others said that they had conversations with friends before:

> '. . .some of them [friends] had kids at a very young age. So, it was mostly their advice that made me think twice about getting myself in such situations, yes.' [Grp. 1. Participant 100]

However, one participant narrated that despite her mother being her primary caregiver, she never had any conversations with her about sex:

> 'I didn't talk with mother at all. Looking at the experience that I've had, I wish maybe that she had told me more, told me more about sex.' [Grp. 1. Participant 105]

Some of the reasons provided why conversation never occurred for the two participants included culture and generation gap between adolescents and their caregivers.

> 'In the Black society, it is totally forbidden to speak to the elders about sex.' [Grp. 1. Participant 104]

**Group 2 participants (those who became pregnant before 18 years).** Out of the ten participants in group 2, six reported that they did not have any conversations about sex with any of the primary caregivers. One participant said: 'I would have liked to talk to grandma about sex, maybe I wouldn't have had sex at such an early age.' [Grp. 2. Participant 227].
One participant believed lack of conversation was due to caregiver not knowing what to say or how to start such conversation:

> 'They [caregivers] really they can't say anything to you because they don't know what to say or how to start the topic.' [Grp. 2. Participant 208]

In addition, culture and taboos were mentioned by three participants as other reasons why conversations did not occur–suggesting that their 'culture' prohibited conversations about topics such as sex:

> 'It's just in our culture, but you don't. You just don't talk about sex.' [Grp. 2. Participant 227]

The few who said that they had conversations about sex reported that conversation was initiated when it was late–when they were either already sexually-active, pregnant or after giving birth:

*'I regret a lot of things when I think about my past. She only started the conversation when I was pregnant. I really wish that my mom had taken the time to sit down and talk to me earlier.' [Grp. 2. Participant 234]*

*'She just didn't really think that I was there or that I was thinking about sex. I think she only maybe started mentioning it when I got to university, she didn't know that I had already started having sex.' [Grp. 2. Participant 237]*

Others mentioned that conversations with grandmothers or aunts were too general or untimely and only emerged when adolescents were pregnant or after giving birth. The content of the conversations that occurred were largely about menstruation and condom use. Notably, two participants mentioned that conversations were delivered in a way of warnings or what has traditionally been associated with 'scare tactics' which included phrases like 'boys give you babies' or 'sleeping with a boy will get you pregnant,' as exemplified below:

*'It wasn't like a conversation, they would lecture us by saying, 'this is what happens, if you have sex, you get pregnant!' [Grp. 2. Participant 237]*

For participants who had had conversations before sexual debut, they revealed that the information was helpful, and it guided them on what they should do to avoid early pregnancies. One recurring theme among participants who became pregnant was the desire for their caregivers, particularly their mothers, to have taken the time to talk to them about sex when they were younger. One participant said:

*'I actually regret a lot of things when I think about my past. And I really wish that my mom had taken the time to sit down and talk to me. Even if she didn't like it or didn't like to, I think things would be a whole lot different.' [Grp. 2. Participant 234]*

## Risk factors for early pregnancy

Participants in both groups experienced similar risk of becoming pregnant before 18 years of age.

**Family issues.**   Both groups reported family issues, including strict parents, unstable families (fragmented due to divorce, separation or death of parents), or generally having poor family relationships which hindered conversations about sex. These challenges exposed adolescents to risks of early pregnancies:

*'I only talked to my sister [. . .] when my sister told mother, she was shouting most of the time saying why did I have sex, why didn't we use a condom.'* [Grp. 1. Participant 117]

Women who lived as adolescents with grandparents reported finding it particularly difficult to talk about sex and sexuality due to the generation gap. Some participants thought their elderly grandparents (or parents) were too 'old-fashioned' to talk about sex.

*'I didn't feel comfortable talking to my grandma about it because she is old.' [Grp. 2. Participant 219]*

Limited conversations about sex were also linked to culture and taboos around such topics. For example, some participants felt that talking about sex would signify being rude to their parents or would signify that one was already engaging in sex. Others mentioned that conversations about sex were perceived by their caregivers as a way of encouraging sex:

*'If they [caregivers] talk about it, it seems like they are promoting it.' [Grp. 1. Participant 104]*

**Peer/friends influence.** Both groups discussed being exposed as adolescents to a great deal of negative peer pressure (both boys and girls) surrounding sex and sexual behavior. Some reported that older friends who were sexually experienced tended to have a negative influence, and often encouraged them to have sex even if participants indicated that they were not yet ready. One common trend that was discussed among participants in both groups was the pressure from friends and peers to have sex with one's boyfriends:

*"My friend said, 'if you aren't having sex with your partner, it means that your partner is having sex with someone else and not you, so you should just do it.' They would say. It is nice, just try it. . ." [Grp. 1. Participant 101]*

However, one difference between the two groups of participants was the ability to rise above and overcome the peer pressure. It was clear that peer pressure existed, and was particularly strong at times, but participants who did not become pregnant by age 18 years were largely focused on future goals, particularly school completion, and took specific steps including contraceptive use, consistent condom use, or delaying having sex to try and prevent pregnancy. One participant provided a detailed account of how she overcame peer pressure:

*"Like a classmate who used to talk to us and tell us about her stories, yes, about her boyfriend and stuff, and at that time I wasn't even thinking about having sex. She would tell us 'it is nice, you should try it'. I said, 'no I don't want babies.'" [Grp. 1. Participant 101]*

**Religion.** Religion was also discussed as a risk factor in the sense that, sex topics were rarely discussed in church and adolescents who were found engaging in sex were considered sinners. Ultimately, this kind of perception prevented adolescents from discussing sexual issues that were bothering them with members of their faith group:

*'Well in church you can't really talk about sex because everyone in church is holy. . .' [Grp. 2. Participant 208]*

*'At church, they always preach that sex before marriage is a sin. So, if you were doing it, you don't feel comfortable talking about sex.' [Grp. 2. Participant 219]*

### Protective factors against early pregnancy

Despite being sexually-active or exposed to several risk factors, more participants from group 1 demonstrated ability to navigate choices such as consistent condom use to prevent early pregnancies and desire to finish school. Family support was key and largely centered on timely communication or conversations about sex and providing general guidance to adolescents. Other social support structures within their community including school, health services and peers were found to be paramount in helping to protect adolescents from early pregnancies as summarized below:

**Adolescents' ability to navigate choices.** The most frequent action mentioned by adolescents who successfully prevented early pregnancy was consistent condom use. While other

methods of protections such as contraception pills were cited, condom use was mentioned as a primary action by almost all the participants who did not become pregnant by age 18 years. The ability to use protection depended on several factors. One was the ability to communicate with one's partner and their partner's receptivity to using condoms. Participants who indicated that they had an understanding partner, or had been in a long term relationship with one partner (as opposed to multiple partners) discussed how helpful this was in negotiating condom use and preventing early pregnancies:

> *'Yeah, I said [to partner] that we should use condoms and he was very supportive.' [Grp. 1. Participant 104]*

> *'Yes, I only had one partner. We were in a long term relationship, and he agreed to using condoms until we reached 18 years.' [Grp. 1. Participant 103]*

In addition, participants who had prior knowledge about various forms of contraception from their education or conversations with friends, family members, or clinic staff, were more aware of the ways in which they could protect themselves:

> *'I mean I knew about fertility and stuff I mean when I was like 9, by the time I became a teenager I was well aware of sex and yeah.' [Grp. 1. Participant 109]*

Moreover, adolescents' ability to seek additional information and make decisive choices were said to also protect them from early pregnancies as exemplified below:

> *'It really had nothing to do with her [Aunt], because its up to me to choose, so yeah I decided to use condoms'.* [Grp. 2. Participant 208]

**Family support.** As discussed above, family was a key pillar in supporting adolescents during their adolescence. Adolescents who mentioned having a good relationship at home, also revealed that their caregivers introduced conversations about sex in good time. One participant said: *'Like I think that I was fortunate enough that I was in a very open family where sex wasn't taboo just to speak about it, it wasn't anything foreign or anything.' [Grp. 1 Participant 109]*

**Social support systems.** Other supportive social systems such as school, peers/friends, health services and religion were indicated to be key in helping adolescents navigate through the decisions they made, and thus helped them to prevent early pregnancies. First, school through life orientation courses–programs that are taught in all schools in South Africa and contains a component that addresses health and sex education, were found to be key in supporting adolescents through providing more information not only about pregnancies but also about sexually transmitted diseases; this communication rarely happened at home. One participant said:

> *'The teachers taught us everything about sex. They taught us about, uhm, all those menstruations, STDs and everything'. [Grp.1. Participant 114]*

Although life orientation courses were provided to all students (both boys and girls), the manner in which the courses were presented to students influenced their perceptions towards the information they received:

*'Talking about it was a big issue, we all didn't know anything about all these things. So, it was all just said in a cruel manner. Yah, it was so awkward, but we listened.'* [Grp. 1. Participant 105]

Two participants from group 2 revealed that such courses were presented in a way to scare adolescents as exemplified below:

'*Well, they [teachers] told us about sexually transmitted diseases. Most of the time I think they talked about that because, the teacher wanted to scare us.'* [Grp. 2. Participant 219]

In addition, participants in both groups reported that the courses were not detailed–they recalled that the courses focused more on condoms, contraception, menstruation and less on STIs and HIV/AIDS:

'*Yes, we did [get life orientation course] but not that deeply. They used to teach us about menstruation but not going deeply.'* [Grp. 2. Participant 233]

In this context, integrating multiple sources of information as discussed in this paper, such as support from home, religion, and school were key in enhancing ways in which adolescents protected themselves against early pregnancies.

Second, exposure to friends and peers who were also intent to avoid pregnancy helped adolescents in decision making and positive choices that protected them from an unintended early pregnancy:

'*We talked about the disadvantages of having sex in high school, and, whether you can get the infections or get pregnant.'* [Grp. 1. Participant 101]

Most participants mentioned that they were more comfortable talking to their peers or friends compared to older people:

'*It was comfortable talking with my friends. They could talk freely about condoms flavors and so on.'* [Grp. 1. Participant 117]

Third, a few participants also mentioned that health care staff at the clinic were helpful in providing them information and contraceptives that empowered them to prevent early pregnancy:

'*Well when we got there, we were actually looking for information about STDs and STIs, yeah, so the nurse was telling us the different types of STIs we should look out for, how it is contracted, yeah.'* [Grp. 1. Participant 104]

Table 2 provides a summary of themes and illustrative excerpts on factors that protected against pregnancy before 18 years of age.

## Discussion

The novelty of this study is that we utilized longitudinal data from adolescence to young adulthood to select participants that were sexually active in adolescence to understand the risks they were exposed to yet did not become pregnant. Interviews with the two groups of women– those who did not become pregnant before 18 years and those who became pregnant before 18

**Table 2. Summary of protective factors against early pregnancy.**

| Protective factors | | quotations |
|---|---|---|
| **Family support** | This includes living with mother, and lessons gained from family members. | '[living with mother] I mean I knew about fertility and stuff when I was like 9, by the time I became a teenager I was well aware of sex.' [Grp. 1. Participant 109] |
| | | 'Also, them [sisters] having kids at a very young age, so it was more like advising me.' [Grp. 1. Participant 100] |
| **Ability to make informed choices & Access to information.** | This included consistent condom use; adolescent initiated condom use. | 'I've always had condoms with me every time. They are in my toiletry bag, my purse, always have them.' [Grp. 1. Participant 117] |
| | | 'I went out there looking for information without anyone telling me to look for information.' [Grp. 1. Participant 119] |
| **School** | Motivation to complete school, life orientation courses at school. | 'Well, I think they [school mates] made me open my eyes at the time, because I had seem most of the people in high school, they hadn't finished metric. So all I wanted to do was finish high school.' [Grp. 1. Participant 101] |
| | | 'Uhm they [teachers] taught us everything about sex.' [Grp. 1. Participant 114] |
| **Friends/peers** | Exposure to good friends/peers | 'Yes, we all advised each other that it is best to use condoms because we were still at school' [Grp. 1. Participant 100] |
| | | 'He [partner] was a little bit older than me so he knew everything you know. So, he was very cautious in talking me through it.' [Grp. 1. Participant 109] |
| **Hospital** | Access to the clinic | 'I used to attend the Birth to Twenty clinic, and they would tell us everything about sex [. . .].' [Grp. 1. Participant 101] |

years–enabled us to understand how some adolescents, living in an urban context and who are exposed to similar multiple risks of becoming pregnant, are able prevent unplanned pregnancies. The key themes that emerged as protective factors against adolescent pregnancies were: (i) adolescent's ability to navigate choices and have future- oriented goals coupled with a supportive environment to achieve these goals; (ii) family support and open conversations about sex; and (iii) social support, in particular school and healthcare services.

Similar to other studies, participants in this study reported having been exposed to multiple risks, such as negative influence from friends/peers who pressured them to have sex, culture/taboos and religious beliefs that prohibited communication about sex amongst young people [12]. Individual related factors–multiple sexual partners, improper use of condoms and use of alcohol and drug abuse, were also discussed as risk factors for sexual activities [7]. Despite this, our findings suggest that adolescents who did not become pregnant by 18 years were able to navigate choices, distinguished between healthy and unhealthy behaviors, and integrated multiple sources of information to protect themselves. This was enhanced by having supportive family relationships and positive peers [15]. Other studies have attributed adolescents' decision making capacity not only to their social support systems [19], but also, to their stage of neuro-development, including complex reasoning [29]. In addition, although adolescents are prompted to make decisions based on immediate rewards, we found that adolescents were also motivated to make positive choices regarding future benefits, for example, their ability to finish school and live a better life [30].

The family was key in protecting adolescents against unplanned pregnancies. Having a supportive family with good communication was key in having open conversations about sex and learning about various pregnancy preventive measures. Conversely, many women who became pregnant before 18 years reported that they came from unstable families with poor relationships at home, which diminished possibilities of conversations about sex before sexual debut. Our findings are in line with previous studies which have reported that parents are key social actors in advising adolescents on matters relating to sex [31]. Previous studies from South Africa have attributed adolescents' risky sexual behaviors to a lack of parental supervision, partly due to dysfunctional families and poor mother-daughter relationships [20]. In addition, recent studies from Soweto have shown that adolescents who do not live with both biological parents are at increased risk of adverse sexual outcomes [14]. Importantly, even with presence of both parents, some participants mentioned lack of conversations about sex, or instances where conversations were delivered in a 'warning' tone, than to constructively provide knowledge [32] and discuss choices and options [33]. Parents need to be supported and enlightened on how to introduce sex related topics to adolescents.

Many participants who did not become pregnant before 18 years revealed how they were supported by their peers in making decisions to finish their studies and protect themselves from risky sexual behaviors (such as through consistent condom use). In addition, some participants reported that being in a long term relationship and having a single partner was beneficial as it enhanced healthy discussions with their partner on pregnancy preventive measures including consistent condom use. Earlier studies from South Africa have shown that girls or young women who have a single partner, or those in a long term relationships are less likely to get unintended pregnancies due to their ability to negotiate condom use [15]. Importantly, our findings show that exposure to positive peers plays a critical role in the lives of adolescents by serving as support for one another, and as trusted sources of information. Peer-to-peer support can be encouraged through building knowledge about sexual health issues amongst youth and providing opportunities to practice intrapersonal/communication skills that can facilitate effective interactions and support.

Schooling was found to be a key motivating and protective factor against early pregnancy. Although women were interviewed when they were 24 years old, those who did not become pregnant before 18 years revealed that they were motivated to finish secondary school in order to achieve a better future life. Others indicated that life orientation lessons provided at school were informative and helped them in their decision making. However, our findings demonstrated that participants who had other support structures (such as from caregivers, peers) were able to integrate what they were taught at school better, compared to those who relied solely on school support. Indeed, our findings illuminate the positive aspect of school environments and calls for education professionals to ensure that life orientation courses are detailed to cover key issues that will form a good foundation of sexual health knowledge at schools. Schools can also work closely with health services, such as through the Integrated School Health Programme (ISHP) in South Africa [9] to ensure that adolescent girls and boys have access to health information and other support that may help them prevent risky sexual behaviors.

Despite the potential role of health services in the provision of reproductive health services to women, only a few participants mentioned accessing care at the clinics in Soweto. This study did not explore further on issues around access to healthcare facilities in Soweto and this warrants further exploration. However, recent studies in South Africa have postulated that many women continue to face challenges accessing reproductive health services, including cost of care, transport to hospital or poor services at the hospital [34]. Thus, it is important to strengthen healthcare systems to offer adolescent-friendly reproductive health services in

South Africa [35] and other similar settings. Given the link between various contextual factors and their influence on risk/protection to adolescent pregnancies, future studies should employ ethnographic approaches in investigating this topic, as this may provide further insights and understanding around adolescent conversations about sex.

## Implications

We found that when supported, adolescents can make life-important choices pertaining to their health and wellbeing, including preventing unwanted pregnancies. As such, policy makers should work to support adolescents through sex education and access to healthcare services and health information to help them make good choices in their lives, and this aligns with Target 3.7 of the SDGs which calls on countries "by 2030, to ensure universal access to sexual and reproductive healthcare services, including for family planning, information and education, and the integration of reproductive health into national strategies and programmes [36]. This is particularly important given that the choices that adolescents make have consequences in terms of their health and well-being during adulthood [37].

Family plays a crucial role in shaping adolescents' decisions. Policy makers should devise strategies to ensure that parents and other caregivers are informed about adolescent health, helping caregivers to see the value in conversations about sex with their children, introducing conversations on time and appreciating their lasting impact in improving adolescent health now and in the future. This can be an effective way to ameliorate the risks reported in this study and can set the stage for future open communication between parents and children.

South Africa has a range of well-established policies that aim to address various determinants of adolescent pregnancy. These includes the youth and adolescent health policy, which aim to strengthen and provide guidance to efforts aimed at both preventing and responding to health needs of young people [38]; the national contraception and fertility planning policy aimed at reprioritizing contraception and fertility planning in South Africa [39] among others. Policy makers should build on such existing policies by ensuring that these policies are centered around adolescents themselves; and future interventions should be framed in ways that are fundamentally empowering for young women.

This study builds on previous studies that have explored risks and protective factors against unwanted pregnancy amongst adolescents. Our findings highlight the importance of using multi-sectoral approaches- focusing on the potential of family, schools, hospitals, and peers in promoting and protecting adolescent health. There is need for future studies to include adolescent partners in understanding risks and protective factors to unwanted pregnancies in South Africa.

## Limitations

A key limitation to this study is that we did not interview adolescent's partners (for both groups), which could have provided more insights on protective factors or barriers to prevent adolescent pregnancies. Future research should aim to address this gap. Secondly, the results provided here might have been affected by recall bias. Interviews were conducted when participants were 24 years old, when they were asked about pregnancy prevention efforts during adolescence, events that occurred as much as 10 years earlier. It is possible that participants might have forgotten some information that is important to understanding their experiences as adolescents. In addition, their reflections on these factors may differ from what it was like when they were younger. However, most participants were able to talk a great deal about their efforts, conversations they had with peers and family members, and steps they took to prevent pregnancy.

## Conclusion

In conclusion, the findings highlight that for complex social challenges, such as unplanned adolescent pregnancies, a multifaceted approach of community efforts to support parents to better engage with their children around life choices and sex, and enabling schools and health services to deliver relevant information in an accessible and acceptable way is paramount. We understand that knowledge alone is not enough to impact adolescent choices and behaviors, and consequently, innovative interventions to boost adolescent agency to stay and complete school education and build supportive peer networks are also essential components of a multi-faceted approach.

## Supporting information

**S1 File. Interview guide.** This qualitative interview guide shows the questions that guided the interviews in this study.
(DOCX)

## Acknowledgments

We acknowledge all participants who took part in this study and colleagues at SAMRC Developmental Pathways for Health Research Unit for their guidance and input during this project.

## Author Contributions

**Conceptualization:** Katharine Chiseri, Dawn L. Comeau, Linda Richter, Aryeh D. Stein, Shane A. Norris.

**Data curation:** Katharine Chiseri.

**Formal analysis:** Edna N. Bosire, Katharine Chiseri, Dawn L. Comeau, Linda Richter.

**Funding acquisition:** Shane A. Norris.

**Investigation:** Aryeh D. Stein, Shane A. Norris.

**Methodology:** Edna N. Bosire, Katharine Chiseri, Dawn L. Comeau, Linda Richter, Shane A. Norris.

**Supervision:** Aryeh D. Stein, Shane A. Norris.

**Writing – original draft:** Edna N. Bosire.

**Writing – review & editing:** Edna N. Bosire, Katharine Chiseri, Dawn L. Comeau, Linda Richter, Aryeh D. Stein, Shane A. Norris.

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
