## [Decision Letter · Decision Letter 0]

23 Sep 2021

 PGPH-D-21-00401 A qualitative study of risks and protective factors against pregnancy among sexually-active teens in Soweto, South Africa. PLOS Global Public Health

Dear Dr. Bosire,

Thank you for submitting your manuscript to PLOS Global Public Health. After careful consideration, we feel that it has merit but does not fully meet PLOS Global Public Health’s publication criteria as it currently stands. Therefore, we invite you to submit a revised version of the manuscript that addresses the points raised during the review process.

We look forward to receiving your revised manuscript.

Kind regards,

Hannah Tappis, DrPH, MPH

Academic Editor

Journal Requirements:

1. In the online submission form, you indicated that "The data that support the findings of this study are available from the corresponding author on reasonable request."

2. Please amend your detailed Financial Disclosure statement. This is published with the article, therefore should be completed in full sentences and contain the exact wording you wish to be published.

i) State what role the funders took in the study. If the funders had no role in your study, please state: “The funders had no role in study design, data collection and analysis, decision to publish, or preparation of the manuscript.”

Additional Editor Comments (if provided): This is a strong research article on a topic of great importance. Please consider the constructive comments below as questions as suggestions to strengthen the manuscript, at your discretion.

Reviewers' comments:

Reviewer's Responses to Questions

**Comments to the Author**

1. Does this manuscript meet PLOS Global Public Health’s publication criteria? Is the manuscript technically sound, and do the data support the conclusions? The manuscript must describe methodologically and ethically rigorous research with conclusions that are appropriately drawn based on the data presented.

Reviewer #1: Yes

Reviewer #2: Yes

2. Has the statistical analysis been performed appropriately and rigorously?

Reviewer #1: N/A

Reviewer #2: Yes

3. Have the authors made all data underlying the findings in their manuscript fully available (please refer to the Data Availability Statement at the start of the manuscript PDF file)?

Reviewer #1: Yes

Reviewer #2: Yes

4. Is the manuscript presented in an intelligible fashion and written in standard English?

Reviewer #1: Yes

Reviewer #2: Yes

5. Review Comments to the Author

Reviewer #1: This is such a well-written paper, engaging and scientifically sound! I wish to congratulate the authors for the effort put into producing this excellent manuscript, the methods are thoroughly described and yet concise. I noted that interviews were conducted in English- was this "piloted" first to ensure the young women completely understood the questions? I'm asking this because many times, young people would say they are comfortable with English but only to prevent being embarrassed and shamed of being unable to understand or speak English... I also noted that there was a "multi-lingual" research assistant present during the interviews- was this the note taker? it would be good to see some instances where young women expressed themselves in their own language other than English to illustrate the freedom they had to express themselves in their own language...

Otherwise, this a very good and timely paper and I completely agree with authors that family institutions are and should be considered as the primary source of information for young people and thus engaging parents/caregivers in pregnancy prevention interventions is the necessary step. Congratulations again!

Reviewer #2: A qualitative study of risks and protective factors against pregnancy among sexually active

teens in Soweto, South Africa.

This manuscript addresses a hugely important topic within South Africa and gives insight into the risk and protective factors of adolescent pregnancy. Overall, this manuscript is well-written and makes a considerable contribution to the field. I have a few minor suggestions below.

Introduction

Line 77 – which SSA countries are you referring to? This might be important for context.

Could you add a reference for early sexual debut being defined as being before 15 years of age or add a comment/explanation in the methodology, please?

A small discussion re. some of the risks of adolescent pregnancy for adolescent mothers and their children i.e. school dropout, HIV transmission, poor birth outcomes, development outcomes for children would be useful within the introduction for contextual purposes – this could also be linked to the SDGs if useful to your overall framing.

Methods

Methods are clear.

Given the sensitive nature of the interview guide, please could you add a note on how confidentiality was maintained during the data collection process.

I appreciate the section on reflexivity – this is hugely important for contextualising results.

Results

Section 1 – sexual activity and risk: A note that this is amalgamated findings forum group 1 and group 2 would help the reader regarding the clarity of this theme.

Do you have data on how many children the mothers in the sample had – it might be interesting to explore whether any of these risk and protective factors related to multiple pregnancies during adolescence? Or if this was even discussed?

Discussion

Thought provoking and thoughtful. Implications of results for policy programming and future studies are clear. A link back to SDGs and future outcomes for adolescents may help strengthen this further but this is just a suggestion.

An additional note in the limitations section further highlighting the impacts of having a sample which is currently 24 years of age rather than adolescents would be useful for clarity.

Conclusion

You refer to teenage pregnancy here and in the title. In the remainder of the manuscript, you focus on adolescent pregnancy. For clarity it may be useful to only use one term.

6. PLOS authors have the option to publish the peer review history of their article (what does this mean?). If published, this will include your full peer review and any attached files.

**Do you want your identity to be public for this peer review?** For information about this choice, including consent withdrawal, please see our Privacy Policy.

Reviewer #1: **Yes: **Kim Jonas, PhD

Reviewer #2: **Yes: **Kathryn J. Roberts

---

## [Editor Report · Decision Letter 1]

18 Oct 2021

A qualitative study of risks and protective factors against pregnancy among sexually-active adolescents in Soweto, South Africa.

PGPH-D-21-00401R1

Dear Dr. Bosire,

We're pleased to inform you that your manuscript has been judged scientifically suitable for publication and will be formally accepted for publication once it meets all outstanding technical requirements.

Within one week, you'll receive an e-mail detailing the required amendments. When these have been addressed, you'll receive a formal acceptance letter and your manuscript will be scheduled for publication.

An invoice for payment will follow shortly after the formal acceptance. To ensure an efficient process, please log into Editorial Manager at https://www.editorialmanager.com/pgph/ click the 'Update My Information' link at the top of the page, and double check that your user information is up-to-date. If you have any billing related questions, please contact our Author Billing department directly at authorbilling@plos.org.

Kind regards,

Hannah Tappis, DrPH, MPH

Academic Editor